# Peel to Flesh Bioactive Compounds Ratio Affect Apple Antioxidant Potential and Cultivar Functional Properties

Monika Sawicka [1], Piotr Latocha [2] and Barbara Łata [1,*]

1 Section of Basic Research in Horticulture, Department of Plant Protection, Institute of Horticultural Sciences, Warsaw University of Life Sciences—SGGW, Nowoursynowska 159, 02-776 Warsaw, Poland
2 Department of Environmental Protection and Dendrology, Institute of Horticultural Sciences, Warsaw University of Life Sciences—SGGW, Nowoursynowska 159, 02-776 Warsaw, Poland
* Correspondence: barbara_lata@sggw.edu.pl

**Abstract:** More than ten thousand apple cultivars with high variability in the quality traits and year-round availability place apples at the forefront of consumed fruits. Yet consumers and producers alike are still looking for new apple cultivars with specific quality attributes such as plant resistance to biotic and abiotic stresses as well as a high health-promoting potential. The model plants were three cultivars: a new 'Chopin' and an old 'Granny Smith', scab resistant cultivars with green peel, and a red-skinned cultivar 'Gala Schniga'. Apple peel and flesh were analyzed separately during two growing seasons: 2016 and 2017. The total ascorbate and phenolics as well as individual phenolic compounds, such as (+)-catechin, (−)-epicatechin, chlorogenic acid, phloridzin, and rutin concentrations, proved to be highly tissue-type and cultivar dependent. The apple of the 'Chopin' and 'Granny Smith' cultivars expressed much lower skin-to-flesh antioxidant potential differences as compared to 'Gala Schniga'. The lowest differences between tissue types were observed in the case of chlorogenic acid and flavan-3-ols, followed by total phenolics and ascorbate concentrations. Except for phloridzin, 'Gala Schniga' exhibited the highest differences in global and individual phenolic compound concentrations as well as total antioxidant capacity between the apple peel and flesh. 'Chopin' was definitely distinguished by the highest concentration of ascorbate in both the peel and the flesh and expressed a higher concentration of flavanols, especially compared to 'Granny Smith'. In contrast to 'Gala Schniga', 'Chopin' was richer in chlorogenic acid, (+)-catechin, and (−)-epicatechin in the flesh. The total antioxidant capacity of the green-peel apple cultivars was similar to that of the red-peel one. A narrower range of differences between the concentration of antioxidants in apple peel and flesh could mean better health-promoting properties and might be related to a greater resistance to environmental stress factors.

**Keywords:** *Malus domestica* borkh.; flavan-3-ols; rutin; chlorogenic acid; ascorbate; trolox equivalent antioxidant capacity; antioxidant distribution pattern

## 1. Introduction

Fruit attributes like size, colour, and no injuries on the skin are considered the main traits that most consumers and producers pay attention to. However, as scientists point out, the new studies should also be focused on internal quality and show the nutritional values to the consumers [1]. Apple fruit internal quality is characterised by a great number of biologically active compounds, such as phenolic compounds, ascorbic acid, thiols, β-carotene, phytosterols, and dietary fiber including pectins [2–5]. In the phenolic compounds group, chlorogenic acid, (+)-catechin, (−)-epicatechin, rutin, phloridzin, quercetin, and a group of procyanidins are found in the highest concentration in apple fruits [6–9]. Most of these compounds exhibit antioxidant activity and are involved in the free radical removals, the excess of which (oxidative stress) can lead to serious damage to important cell components like nucleic acids, proteins, or lipids [10]. The health benefits of apples and other fruits in

the prevention of many diseases, including cardiovascular diseases, cancers, and diabetes, have been considered in several studies [11–13]. By measuring the tissue's total antioxidant activity/capacity, it is possible, to some extent, to summarize the biological activity of many compounds and/or the health benefit potential of the tested material, including foods [14]. As with any other fruits, apple phytochemicals are mainly concentrated in the epidermal zone (peel), which has a protective function against environmental stresses [15–17]. Moreover, the apple peel could be a good indicator of the antioxidant potential of the whole fruit due to the high correlation between apple peel and whole fruit antioxidant concentrations [18]. Considering the fresh weight of the fruit, the peel to flesh antioxidants ratio seems to be of great importance regarding total functional properties per fruit [2]. Usually, red-skin apples have a higher antioxidant capacity than green-skin apples due to anthocyanins accumulation, which affects the global flavonoids concentration—the main subgroup of phenolic compounds [19–21]. However, this group of compounds is absent in the flesh of most cultivars. Apart from the tissue type, the cultivar is the important factor determining apple phytochemicals concentration and distribution [5,22–24]. More than ten thousand apple cultivars, as indicated by the European Apple Inventory, with high variability in the quality traits and round availability, place apples at the forefront of consumed fruits [1,22]. Both consumers and producers are looking for new cultivars of apples, although each of these groups may initially emphasize different quality determinants. Consequently, their expectations are similar because these cultivars that express a high concentration of biologically active compounds show, as a rule, greater resistance to biotic and abiotic stress during the growing period [10,16,25,26]. The new varieties selected by high-quality standards can gain a competitive advantage on the fresh fruit market [27].

Antioxidants are synthesized in plant tissues in response to excess light, drought, extreme temperatures, excess or deficiency of soil essential nutrients, or by a pathogen or pest attack [10,15,25]. Therefore, the accumulation of antioxidants is highly dependent on the weather and soil properties during the growing season, which is directly or indirectly involved in stress induction [22,23,28]. Increased plant resistance, resulting from the high efficiency of the antioxidant apparatus, gives the opportunity to cultivate the plant in a more sustainable way (lower agrochemicals usage), thus reducing the risk to humans and the environment [29]. Reducing the use of agrochemicals in agriculture, as well as increasing the acreage of organic farming, is today one of the priority tasks in the European Green Deal [30–32]. Cultivars characterized by high resistance to biotic and abiotic stress are one of the important elements of this strategy.

The aim of the study was to evaluate the antioxidant properties of the newly developed, and old green-skinned apple 'Chopin' and 'Granny Smith' cultivars, respectively, in relation to the red-skinned 'Gala Schniga'. In relation to the selection of experimental material, we would like to add the main aim of the study was to characterize the new cultivar 'Chopin' with green skin and higher fruit acidity, resistant to biotic stresses. We hypothesized that due to the high resistance of this cultivar to biotic stresses, it will have high antioxidant potential, and it would be valuable to compare it with the red-skinned cultivar, also economically important. There are a few green-skin apple cultivars, most of which are old cultivars with a small cultivation area. Total phenolic and ascorbate concentrations, individual phenolic compound concentrations, as well as total antioxidant capacity, were monitored in the apple peel and flesh during two growing seasons (2016 and 2017), expecting also that peel-to-flesh bioactive compounds ratio can shape apple antioxidant potential, cultivar functional properties, and help to explain some aspects of resistance to environmental stresses.

## 2. Materials and Methods

### 2.1. Plant Material

'Gala Schniga' (Gala S.) and 'Chopin' cultivars were harvested from the Experimental Orchard of the Department of Pomology at Warsaw University of Life Sciences-SGGW in Warsaw, Wilanow, Poland (52°9′36.1″ N 21°5′58.2″ E). The fruit was sampled in 2016

and 2017. Green peel scab-resistant 'Chopin' is a new cultivar created from the crossing of 'Granny Smith' with scab-resistant clone 'U 211' at Warsaw University of Life Sciences in Poland (Author: E. Pitera). Apples of the new cultivar are medium-sized with green peel and flavor and between sweetness and acidity. 'Chopin' is resistant to *Venturia inaequalis* and less vulnerable to *Podosphaera leucotricha* [33]. 'Granny Smith' (Granny S.) cultivar was obtained from the local market because this cultivar does not mature in Poland's climate. The plants in the Wilanow orchard were cultivated in line with Integrated Crop Management.

### 2.2. Sample Preparation

Apples were randomly collected from apple trees in three biological samples. Fruits were equal in size and without skin injures. Apple peel (AP) and flesh (AF) were tested. Flesh and peel with a thin layer (1–2 mm) of flesh were frozen in liquid nitrogen and stored in deep freeze ($-80\ ^\circ$C) until analysis. Chemical analyses for all components were conducted in three replicates for each cultivar, each consisting of 3 fruits taken from one tree. In addition, each measurement was carried out in at least 2 analytical replicates.

### 2.3. Determination of Bioactive Compounds

Total ascorbate concentration, i.e., L-AA + DHAA (the sum of L-ascorbic and dehydroascorbic acids, respectively), has been analysed. Frozen apple tissues were powdered in liquid nitrogen. Extraction was conducted in 0.1 M HCL and PVPP (polyvinylpyrrolidone) powder. Samples were centrifuged at 14 000 rpm at 4 $^\circ$C. To determine total concentration of ascorbate, oxidation of L-AA to DHAA was made by ascorbate oxidase. Next, o-phenylendiamine was added to obtain fluorescent derivatives. Total ascorbate concentration was measured fluorometrically by HPLC ($\lambda$ 350/450 nm) with a Waters 474 Scanning Fluorescence Detector (Waters Co., Milford, MA, USA) using SpherisorbR column for analysis. Measurement was conducted under isocratic conditions using 800 mmol $KH_2PO_4$ buffer solution in 20% methanol (pH 7.8). The results were calculated using an external standard method. More details are described in previously published work [22].

Total phenolic compounds concentration was determined by two methods: using Fast Blue BB (4-benzoylamino-2,5-dimethoxybenzenediazonium chloride hemi [zinc chloride] salt, FBBB) or Folin reagent (for the simplicity in the text we use the terms "FBBB method" and "Folin method") [34]. Tested tissues (AP, AF) were extracted twice in 70% ethanol solution at ultrasonic bath for 30 min at room temperature. Extracts were diluted with redistilled water, apple peel was diluted in 1:5 and flesh in 1:1 ratio (*v/v*). In the FBBB method, extracts or standards (with a volume of 1000 μL) were transferred to spectrophotometric cuvettes, then FBBB (100 μL 0.1% FBBB) and NaOH (100 μL 5% NaOH) were added. After one hour of incubation, absorption of obtained solutions was measured at 420 nm (spectrophotometer UV-VIS (U-2900) from HITACHI (purchased from Dynamica Sci. Ltd., Milton Keys, UK). In the Folin method, extracts or standards (50 μL) were transferred to test tubes, then 430 μL redistilled water and 20 μL of Folin reagent were added. Reagents were mixed, followed by the 20% $Na_2CO_3$ and 450 μL redistilled water completed. After one hour of incubation, absorbance was read at 725 nm. In both methods, results were calculated using a standard curve and expressed in mg kg$^{-1}$ fresh weight (FW) on gallic acid (GA) equivalents (stock solution = 1 mg GA mL$^{-1}$ and working standard concentrations in the range 0–120 μg GA mL$^{-1}$).

Extracts previously prepared for the determination of the total phenolics concentration were used for the HPLC (Waters Co., Milford, MA, USA, UV-vis detector M2487; System Breeze) separation of individual phenolic compounds. Five phenolic compounds were determined: (−)-epicatechin, (+)-catechin, rutin, phloridzin, and chlorogenic acid. In total 1 mL of samples or standards were transferred to the vials; the sample injection volume was 20 μL. 0.01M phosphoric acid and pure methanol were used as mobile phases. Compounds detection was monitored using UV-VIS detector at 280 nm. Quantification was based on an external standard calibration curve [7].



*2.4. Measurement of Total Antioxidant Capacity (TAC)*

TAC was measured by three methods: the ferric reducing antioxidant power (FRAP), 2,2-diphenyl-1-picrylhydrazyl (DPPH), and 2,2′-azino-bis-3-ethylbenzthiazoline-6-sulphonic acid (ABTS) assays. The results of all three methods were calculated using a calibration curve and presented in mmol TE (Trolox equivalents) on the kg FW. The methods were described in detail in previous works [35–37].

*2.5. Statistical Analysis of the Data*

The obtained results were processed by three-way factorial analysis of variance (ANOVA) using Statistica version 13.0 software (TIBCO Software Inc. (2017), http://statistica.io (accessed on 10 January 2023), Palo Alto, CA, USA). The main effects under consideration were: growing season, tissue type, and cultivar. The homologous groups were specified by Tuckey (HSD) test at a 5% probability level.

### 3. Results and Discussion

*3.1. Statistical Analysis*

The significant influence of all main factors (cultivar, tissue type, growing season) on most of the tested parameters was proven (Table 1). Statistical analysis revealed that the most significant effect was the tissue type (AP vs. AF). It should be pointed out that the cultivar and growing season effects depended, to a large extent, on the examined compound. The F-values varied over a wide range, i.e., from 258 (FRAP) to 2477 (DPPH), from not significant 0.0 (total phenolics FBBB) to 31.6 (ABTS) and from not significant 0.59 (phloridzin) to 119 (ascorbate) for tissue type, growing season, and cultivar, respectively. Cultivar factor has the most impact on (−)-epicatechin and ascorbate concentrations. With a few exceptions, the analysis demonstrated the significance of the interaction between the main factors (Table 1).

**Table 1.** Summary of statistics (test ANOVA) for analysed parameters; F values for particular sources of variation and the level of their significance.

| Parameter | Source of Variation | | | Interactions | | |
|---|---|---|---|---|---|---|
| | Cultivar (A) | Tissue Type (B) | Season (C) | A × B | A × C | B × C |
| df | 2 | 1 | 1 | 2 | 2 | 1 |
| Ascorbate | 119 *** | 649 *** | 10.5 ** | 52.2 *** | 31.1 *** | 9.26 ** |
| (+)-Catechin | 9.10 ** | 322 *** | 11.1 ** | 18.8 *** | 8.73 ** | 4.97 * |
| Chlorogenic acid | 52.4 *** | 889 *** | 0.88 ns | 44.8 *** | 33.0 *** | 121 *** |
| (−)-epicatechin | 236 *** | 1506 *** | 17.8 *** | 284 *** | 28.9 *** | 3.61 ns |
| Phloridzin | 0.59 ns | 545 *** | 2.44 ns | 0.31 ns | 1.63 ns | 4.58 * |
| Rutin | 2.45 ns | n. a. | 16.7 *** | n. a. | 16.5 *** | n. a. |
| Total phenolics FBBB [a] | 11.5 *** | 869 *** | 0.00 ns | 23.3 *** | 9.43 *** | 8.25 ** |
| Total phenolics FOLIN | 2.75 ns | 336 *** | 10.1 ** | 8.91 ** | 7.42 ** | 12.4 ** |
| ABTS [b] | 1.82 ns | 260 *** | 31.6 *** | 6.22 ** | 8.24 ** | 1.97 ns |
| DPPH [c] | 8.46 ** | 2477 *** | 12.9 ** | 0.95 ns | 25.8 *** | 481 *** |
| FRAP [d] | 6.92 ** | 258 *** | 7.21 * | 21.6 *** | 7.09 ** | 3.62 ns |

df—freedom degrees number. *** significant at $\alpha = 0.001$. ** significant at $\alpha = 0.01$. * significant at $\alpha = 0.05$. ns, not significant. n. a., not analysed, rutin was determined only in the apple peel. [a] FBBB, Fast Blue BB 4-benzoylamino-2,5-dimethoxybenzenediazonium chloride hemi [zinc chloride] salt. [b] ABTS, 2,2′-azino-bis (3-ethylbenzothiazoline-6-sulphonic acid). [c] DPPH, 2,2-diphenyl-1-picrylhydrazyl. [d] FRAP, ferric reducing antioxidant power.

*3.2. Biologically Active Compounds in Relation to Cultivar and Tissue Type*

The question about the concentration of phenolic compounds can be classified as the most frequently asked query in the analysis of plant-derived foods [2,6,7,19,20]. This extremely diverse group of secondary metabolites, and one that is involved in numerous functions in plants, are also known for high biological activity. The relationship between (poly) phenol-rich foods and human health has been demonstrated in numerous epidemi-

ological studies [11–13]. Phenolic compounds are characterized by a high antioxidant activity, and in many cases, the quali-quantitative phenolic compounds pattern determines the antioxidant power of fruits or vegetables [25,38]. In this study, 2 methods were used to assess the total phenolic compound concentrations: the commonly used method with Folin's reagent and the method recently developed by Medina [34] using FBBB. The search for a new, simple, and inexpensive method for the determination of the sum of phenolic compounds results from the fact that many non-phenolic compounds often found in fruits and vegetables form blue complexes with Folin's reagent. However, the method utilizing Fast Blue BB diazonium salt is based on the coupling of phenolic compounds with the diazonium salt resulting in the formation of azo complexes and may be more specific [34]. Therefore, we included this method in the analysis of total phenolic compounds concentration in apple fruit to evaluate the result differences between methods.

The results of the statistical analysis differed depending on the method used to determine phenolic compounds. The influence of the cultivar on the phenolic concentration was not proven in the case of the Folin method, and in the case of the FBBB method, the effect of the growing season on the concentration of these compounds was not confirmed (Table 1). Therefore, the method used to measure total phenolic concentrations may give different final results regarding the influence of tested factor (s). The differences in the phenolic concentrations between apple peel and apple pulp ranged from 5.6- (FBBB) to 7.1-fold (Folin) on average (Table 2). Furthermore, the concentration of phenolic compounds was on average about 3 times higher in the apple peel and 3.8 times higher in the pulp, respectively, in the FBBB method opposed to Folin (Table 2). The FBBB method revealed not only a greater concentration of phenolic compounds in general, but also a greater presence of these compounds in the flesh, which narrows the differences between their amount in the epidermal zone and downstream of the epidermal layer. The author of the FBBB method [34], comparing the concentration of phenolic compounds determined by the FBBB and Folin methods in barley and wheat grains, obtained a concentration of phenolic compounds twice as high using a newly developed method. When the research material was that of different types of tea, the differences in phenolic compounds concentration between these two methods were up to 6-fold. The method, however, requires further verification on larger plant samples, along with the evaluation of individual phenolic compounds by HPLC, so that it could be stated with greater certainty that it is more selective in relation to phenolic compounds and allows for a better estimation of their total concentration than the widely used Folin method. Similar conclusions were drawn by other researchers analysing the concentration of phenolic compounds in strawberry fruits [39].

**Table 2.** Effect of the cultivar and tissue type on the total phenolic concentrations (mg GA kg$^{-1}$ FW). Results according to FBBB and Folin-Ciocalteu methods.

| | Method Used | | | | | |
| | FOLIN | | | FBBB [1] | | |
| Cultivar | AP [2] | AF [3] | Av. | AP | AF | Av. |
| --- | --- | --- | --- | --- | --- | --- |
| Chopin | 753 [bc] | 102 [a] | 428 [A] | 2000 [b] | 403 [a] | 1201 [A] |
| Gala S. [4] | 902 [c] | 67.6 [a] | 485 [A] | 2746 [c] | 312 [a] | 1529 [B] |
| Granny S. [5] | 616 [b] | 149 [a] | 383 [A] | 1962 [b] | 482 [a] | 1222 [A] |

[1] FBBB, Fast Blue BB 4-benzoylamino-2,5-dimethoxybenzenediazonium chloride hemi [zinc chloride] salt. [2] Apple peel. [3] Apple flesh. [4] Gala Schniga, [5] Granny Smith. GA, gallic acid equivalent. Mean values (Av.) marked by a different capital letter and mean values within tissue types (AP/AF) denoted by a different lowercase letter separately for each method used, differ significantly at $p \leq 0.05$ (Tuckey's HSD). The presented data are the mean for the two growing seasons of 2016 and 2017 ($n = 6$).

Irrespective of the method used, the flesh phenolic compounds concentration of all tested apple cultivars was similar (Table 2). In turn, the tested apple peel differed considerably in terms of the phenolic concentrations, but the measurement method also influenced the results. Based on the Folin analysis, the 'Chopin' cultivar did not differ

significantly in the concentrations of peel phenolic compounds in relation to the red-skinned 'Gala S.'. According to the FBBB method, the 'Gala S.' cultivar distinctly differed from the other two tested cultivars in terms of the phenolic compounds concentration in the apple peel. In both methods green peel apples 'Granny S.' and 'Chopin' were characterized by similar total phenolic concentrations.

In order to elucidate further differences with respect to phenolic compounds and tissue type, selected individual phenolic compounds were measured using the HPLC technique. The dominant compounds, such as (+)-catechin, (−)-epicatechin, rutin, phloridzin, and chlorogenic acid, were quantified (Table 3). The concentration of individual phenolic compounds did not fully reflect the differences that were obtained with the total concentration of phenolic compounds (Folin vs. FBBB method). Tested cultivars average in two seasons did not differ significantly in the concentration of rutin in the peel and phloridzin, both in the peel and pulp. Among the determined phenolic compounds, rutin was present in the highest concentration, but rutin was detected only in the apple peel. The lowest concentration was recorded for phloridzin irrespective of tested cultivars. Savatovic et al. [40], when determining phenolic compounds similarly to the presented study, recorded the highest concentration of rutin and many times, as much as 26 times lower the concentration of phloridzin. In the other studies [7], (−)-epicatechin, (+)-catechin, and chlorogenic acid were determined approximately 4 times less than the concentration of rutin. In turn, based on the dry weight of the apple peel, the highest concentration was recorded for rutin while the lowest of chlorogenic acid. Taking into account the whole fruit (AP and AF), chlorogenic acid was at the highest concentration and phloridzin at the lowest. However, it should be pointed out that great differences between cultivars were documented [7,23].

**Table 3.** Distribution of individual phenolic compounds and total ascorbate concentrations in the apple peel and the apple flesh depending on the cultivar (mg kg$^{-1}$ FW).

| | Component Tested | | | | | | | | | | | | | | | | |
|---|---|---|---|---|---|---|---|---|---|---|---|---|---|---|---|---|---|
| | Chlorogenic Acid | | | Phloridzin | | | (+)-Catechin | | | (−)-epicatechin | | | Rutin | | | Ascorbate | | |
| Cultivar | AP [1] | AF [2] | Av. | AP | AF | Av. | AP | AF | Av. | AP | AF | Av. | AP | AF | Av. | AP | AF | Av. |
| Chopin | 218 [c] | 79.2 [a] | 149 [B] | 148 [b] | 12.6 [a] | 80.3 [A] | 279 [c] | 72.3 [a] | 176 [B] | 248 [d] | 65.5 [b] | 157 [B] | 499 [a] | n. d. | 499 [A] | 445 [e] | 82.3 [b] | 264 [C] |
| Gala S. [3] | 307 [d] | 76.1 [a] | 191 [C] | 159 [b] | 16.7 [a] | 88.0 [A] | 299 [c] | 36.0 [a] | 168 [B] | 423 [e] | 27.5 [a] | 225 [C] | 638 [a] | n. d. | 638 [A] | 146 [c] | 22.7 [a] | 84.3 [A] |
| Granny S. [4] | 181 [b] | 69.0 [a] | 125 [A] | 149 [b] | 17.5 [a] | 83.2 [A] | 176 [b] | 71.3 [a] | 124 [A] | 113 [c] | 39.3 [ab] | 76.3 [A] | 607 [a] | n. d. | 607 [A] | 277 [d] | 32.1 [ab] | 155 [B] |

[1] Apple peel. [2] Apple flesh. [3] Gala Schniga, [4] Granny Smith. n. d., not detected. Mean values (Av.) marked by a different capital letter and mean values within tissue types (AP/AF) denoted by a different lowercase letter for each component tested separately differ significantly at $p \leq 0.05$ (Tuckey's HSD). The presented data are the mean for the two growing seasons of 2016 and 2017 ($n = 6$).

There were no significant differences between examined cultivars in the chlorogenic acid and (+)-catechin concentrations in apple pulp, but the peel differed significantly. Regarding examined cultivars, 'Gala S.' peel exhibited a significantly higher concentration of chlorogenic acid and (−)-epicatechin compared to 'Chopin' and 'Granny S.', but 'Chopin' was the second highest. For (+)-catechin, peel of 'Chopin' and 'Gala S.' form the same homological group (Table 3). In contrast to other phenolic compounds, the case of (−)-epicatechins is interesting, the concentration of which is significantly different in the tested cultivars, both in the apple peel and in the flesh. Apple flesh of 'Chopin' expressed significantly higher (−)-epicatechin concentrations compared to 'Gala S.'; the opposite situation was for the peel of these cultivars. According to Petkovsek et al., scab-resistant apple cultivars had a significantly higher concentration of some individual and total phenolic concentrations in comparison with the scab susceptible, especially the pulp [16]. In this study 'Chopin' expressed a higher concentration of flavanols, particularly compared to 'Granny S.' In turn, compared to 'Gala S.', 'Chopin' was richer in these compounds in the flesh. Of course, these compounds do not represent all the phenolic compounds identified in apples [41,42], and the issue should be further explored. An important group of apple phenolic compounds are oligomeric flavanols, located primarily in the peel of apples.

In addition, the 'Chopin' cultivar was definitely distinguished by the concentration of ascorbate in both the peel and the flesh (Table 3). Apple peel of 'Chopin' was 1.6 and 3-fold higher while its flesh was 2.6- and 3.6-fold higher in ascorbate concentration compared to 'Granny S.' and 'Gala S.' peels, respectively. The concentration of ascorbic acid in commercial apple cultivars is low compared to other fruits, amounting on average to 10 mg per 100 g fresh fruit weight [43,44]. Since the number of apple cultivars is huge, the variation in ascorbic acid concentration between them can range even between 1.7- and 3.3- in whole apples and may differ nearly 5 times in apple peels [5,18,22]. The share of the peel as a source of ascorbate in relation to the whole apple fruit may amount to even 30%, although its share in the fresh weight of the fruit does not generally exceed 10% [2]. Many external factors influence the metabolism of antioxidants, to which the response of individual cultivars may depend on the genetically defined total antioxidant potential [10,19,21–23,28,45].

Due to a large number of bioactive compounds in the plant tissue, several assays have been developed to help assess total antioxidant activity/capacity [14,35,36]. They are useful in the overall assessment of the health benefit potential of different types of food. Sometimes, more than one method is used when evaluating the antioxidant properties of different food types. This is due to the presence of two groups of antioxidants, hydrophilic, and lipophilic, but within each group, a great number compounds with individual physico-chemical characteristics exist. While the ABTS test can measure both hydrophilic and lipophilic antioxidants, the FRAP method only measures hydrophilic ones while the DPPH test is applicable to the hydrophobic counterparts [14]. However, each of these tests has its limitations with respect to individual compounds; therefore, it is suggested to involve more than one method to assess total antioxidant capacity [46–48]. By evaluating hydrophilic and hydrophobic antioxidants in separate tests, differences in the range of a given group of compounds in the tested samples can be detected. Moreover, because the authors of the studies use different tests, this gives a greater opportunity to compare different study results. In this study, it was decided to measure total antioxidant capacity using three methods—ABTS, DPPH, and FRAP-assays (Table 4). Regardless of the assay test and the cultivar examined, the antioxidant capacity determined for apple flesh did not differ significantly. This confirms the results obtained in the assessment of individual compounds discussed above, where their concentrations in the apple flesh were generally similar. 'Chopin' peel was characterized by the highest total antioxidant capacity based on the DPPH test while 'Gala S.' based on the ABTS and FRAP assays, as compared to the other two cultivars. However, in the case of the FRAP test, both mentioned above cultivars constituted one homologous group. These tests confirm the previously presented results, where 'Chopin' and 'Gala S.' showed alternately the highest ascorbate and/or phenolic compound concentrations. In turn, in the case of the ABTS test, the peel of 'Gala S.' did not differ significantly from the 'Granny S.' one. Although the same results were not obtained for individual tests, it can be concluded that the total antioxidant capacity of the tested green-skinned cultivars did not differ so much from the red-skinned one. The variability in the total antioxidant capacity, both with respect to tested tissue type and cultivar, suggests that further differences between the tested cultivars in the content of an individual, hydrophilic and/or lipophilic, antioxidants can be expected, and the issue can be further studied. The greatest differences between the peel and flesh were noted for the 'Gala S.'. This is likely related to the colour of the peel, indicating a higher concentration of anthocyanins compared to other cultivars. Different groups of antioxidant compounds have different contributions in the total antioxidant capacity. The strongest positive correlation between the total antioxidant activity, and the various compounds, regardless of the test used, was found in the case of phenolic compounds [6,48]. Previous studies have found that phenolic compounds are the main group of antioxidant compounds in apples due to their activity and concentration [23,28]. In this study, the correlation coefficients (the Pearson correlation analysis) between the total antioxidant capacity and the concentration of phenolic compounds were in the range of 0.84 (DPPH) to 0.96 (FRAP) for FBBB method

and 0.79 (ABTS) to 0.92 (FRAP) in case of Folin method. For ascorbate, the values were lower and within the range 0.55 (FRAP, ABTS) to 0.72 (DPPH).

**Table 4.** Total antioxidant capacity in the apple peel and the apple flesh depending on cultivar (mmol TE kg$^{-1}$ FW).

| | Antioxidant Capacity Test | | | | | | | | |
|---|---|---|---|---|---|---|---|---|---|
| | ABTS[5] | | | DPPH[6] | | | FRAP[7] | | |
| Cultivar | AP[1] | AF[2] | Av. | AP | AF | Av. | AP | AF | Av. |
| Chopin | 25.6 [b] | 9.47 [a] | 17.5 [A] | 45.2 [c] | 11.8 [a] | 28.5 [B] | 16.6 [bc] | 5.68 [a] | 11.2 [A] |
| Gala S. [3] | 32.4 [c] | 7.80 [a] | 20.1 [A] | 41.1 [a] | 9.77 [a] | 25.4 [A] | 25.1 [c] | 4.05 [a] | 14.6 [B] |
| Granny S. [4] | 27.4 [bc] | 11.7 [a] | 19.6 [A] | 41.9 [ac] | 10.0 [a] | 26.0 [A] | 15.6 [b] | 7.27 [a] | 11.4 [A] |

[1] Apple peel. [2] Apple flesh. [3] Gala Schniga, [4] Granny Smith. [5] ABTS, 2,2'-azino-bis (3-ethylbenzothiazoline-6-sulphonic acid). [6] DPPH, 2,2-diphenyl-1-picrylhydrazyl. [7] FRAP, ferric reducing antioxidant power. TE, trolox equivalents. Mean values (Av.) marked by a different capital letter and mean values within tissue types (AP/AF) denoted by a different lowercase letter for each component tested separately differ significantly at $p \leq 0.05$ (Tuckey's HSD). The presented data are the mean for the two growing seasons of 2016 and 2017 ($n = 6$).

Tissue type was the most prominent factor that affected total ascorbate, total phenolic compounds, and individual phenolic compounds, such as (+)-catechin, (−)-epicatechin, rutin, phloridzin, and chlorogenic acid, as well as total antioxidant capacity (Tables 1–5). For all compounds, the differences in concentrations between apple peel and flesh were statistically significant. Apple peel was the richest source of bioactive compounds in all varieties. This is the expected result since similar conclusions can be found in other related studies where a more large number of cultivars were tested [18,23,43]. Table 5 summarizes the range of the differences regarding antioxidant concentration between the peel and the flesh of tested apple cultivars. Differences between the peel and flesh in the concentration of the compound strongly depended not only on the cultivar, but also on the type of compoun. The size of the differences between the apple peel and the flesh concentration can be important in terms of the whole fruit compound content. In general, apple fruit of 'Chopin' and 'Granny S.' cultivars expressed much lower skin-to-flesh antioxidant potential differences than 'Gala S.'. Fruits of 'Gala S.' and 'Chopin' cultivars were characterized by similar average fresh weight: 180 and 183 g per fruit, respectively. Granny S. was characterized by a bigger apple, i.e., approximately 224 g per fruit.

**Table 5.** Range of differences between the apple peel and flesh (AP/AF) depending on the cultivar and parameter tested.

| | Cultivar | | |
|---|---|---|---|
| Parameter | Chopin | Gala S. | Granny S. |
| Ascorbate | 5.41 | 6.43 | 8.63 |
| (+)-Catechin | 3.86 | 8.31 | 2.47 |
| Chlorogenic acid | 2.75 | 4.03 | 2.62 |
| (−)-epicatechin | 3.79 | 15.4 | 2.88 |
| Phloridzin | 11.7 | 9.52 | 8.51 |
| Rutin | n. d. | n. d. | n. d. |
| Total phenolics FBBB [a] | 4.96 | 8.80 | 4.07 |
| Total phenolics FOLIN | 7.38 | 13.3 | 4.13 |
| ABTS [b] | 2.70 | 4.15 | 2.34 |
| DPPH [c] | 3.83 | 4.21 | 4.19 |
| FRAP [d] | 2.92 | 6.20 | 2.27 |

AP/AF: -values obtained by dividing the concentration in the peel by the concentration in the flesh. [a] FBBB, Fast Blue BB 4-benzoylamino-2,5-dimethoxybenzenediazonium chloride hemi [zinc chloride] salt. [b] ABTS, 2,2'-azino-bis (3-ethylbenzothiazoline-6-sulphonic acid). [c] DPPH, 2,2-diphenyl-1-picrylhydrazyl. [d] FRAP, ferric reducing antioxidant power. n. d., not detected in the apple flesh.

The lowest differences between tissue types were found in chlorogenic acid and flavan-3-ols, followed by total phenolic compounds and ascorbate. Except for phloridzin, 'Gala S.' exhibited the highest differences in global and individual phenolic compound concentrations as well as total antioxidant capacity between the apple peel and flesh. The range of differences between the peel and the pulp in the antioxidant concentration varied widely from 2.47 ((+)-catechin, Granny S.) to 15.4 ((−)-epicatechin, Gala S.) fold variation (Table 5). The differences in the total antioxidant capacity between the tested tissues were characterized by a narrower range, i.e., 2.27 (FRAP, 'Granny S.') to 6.2 (FRAP, 'Gala S.') fold variation. In studies on a larger material, where 19 cultivars of apples were analysed but the peel and the whole fruit were tested [18], the differences were smaller. A different pattern of compound distribution may indicate a different role of individual antioxidants under stress conditions. Enzymatic and non-enzymatic antioxidants are involved in several processes keeping level of active oxygen species, which are generated during biotic and abiotic stress as well as normal metabolic processes, under control [10,49]. It is expected that cultivars with a more efficient antioxidant apparatus (antioxidant concentration/regeneration or activity) will be more resistant to stress. Simultaneously, a higher concentration of biologically active compounds increases the health-promoting value of the fruit.

### 3.3. Biologically Active Compounds in Relation to the Growing Season

The growing season effect has been proven for most of the tested components (Table 1). The mean monthly temperature and precipitation in 2016 were similar to the long-term average (Figure 1). The next season was more varied regarding these parameters. Mean month precipitation and temperature in 2017 were much higher as compared to long-term averages; 2017 was a hot year with lots of rainfall during the growing season. The differences in weather conditions were probably one of the reasons of high variation in the concentration of bioactive compounds through the tested growing seasons. The growing season effect occurrence is the rule in long-term studies, especially in recent years with observed climate changes [18,31]. Looking at the average levels of examined antioxidants, it can be concluded that 2016 was favorable for their accumulation in apple fruits compared to 2017 (Table 6). In 2016, significantly higher concentrations were recorded in the case of ascorbate, (+)-catechins, (−)-epicatechins, and total antioxidant activity measured by the FRAP and the ABTS tests. In contrast, in 2017, significantly higher concentration of rutin and total antioxidant capacity measured with the DPPH test was revealed. As previously indicated, the influence of the growing season on total phenolic concentrations depended on the measurement method used (Table 6). The antioxidant properties variability in subsequent years ranged from 0% (chlorogenic acid, total phenolics FBBB) to 40% (ABTS). Based on the data from these and other studies [18,22,31], it can be concluded that the weather conditions in the season can significantly affect the internal quality of fruit. The so-called 'year effect' usually consists of many variables (weather components, soil conditions); therefore, it is not easy to clearly assess whether the content of a certain compound(s) was influenced by one of them or by the interaction of many factors.

**Table 6.** The influence of the growing season on the concentrations of biologically active compounds (mg kg$^{-1}$ FW) and the total antioxidant capacity (mmol TE kg$^{-1}$ FW).

| Parameter | Growing Season | | |
| | 2016 | 2017 | 2016/2017 [A] |
| --- | --- | --- | --- |
| Ascorbate | 183 [b] | 152 [a] | 1.2 |
| (+)-Catechin | 173 [b] | 138 [a] | 1.3 |
| Chlorogenic acid | 152 [a] | 158 [a] | 1.0 |
| (−)-epicatechin | 165 [b] | 141 [a] | 1.2 |
| Phloridzin | 88.0 [a] | 79.0 [a] | 1.1 |
| Rutin | 471 [a] | 691 [b] | 0.7 |

**Table 6.** *Cont.*

| | Growing Season | | |
|---|---|---|---|
| Parameter | 2016 | 2017 | 2016/2017 [A] |
| Total phenolics FBBB [B] | 1318 [a] | 1317 [a] | 1.0 |
| Total phenolics FOLIN | 375 [a] | 488 [b] | 0.8 |
| ABTS [C] | 22.4 [b] | 15.8 [a] | 1.4 |
| DPPH [D] | 25.5 [a] | 27.8 [b] | 0.9 |
| FRAP [E] | 13.5 [b] | 11.3 [a] | 1.2 |

[A] 2016/2017 results from dividing the concentration in 2016 by the concentration in 2017. [B] FBBB, Fast Blue BB 4-benzoylamino-2,5-dimethoxybenzenediazonium chloride hemi [zinc chloride] salt. [C] ABTS, 2,2′-azino-bis (3-ethylbenzothiazoline-6-sulphonic acid). [D] DPPH, 2,2-diphenyl-1-picrylhydrazyl. [E] FRAP, ferric reducing antioxidant power; TE—Trolox equivalent; Values in rows marked with a different lowercase letter differ significantly at $p \leq 0.05$ (Tuckey's HSD). The presented data are the mean for the all tested cultivars and type of tissue ($n = 18$).

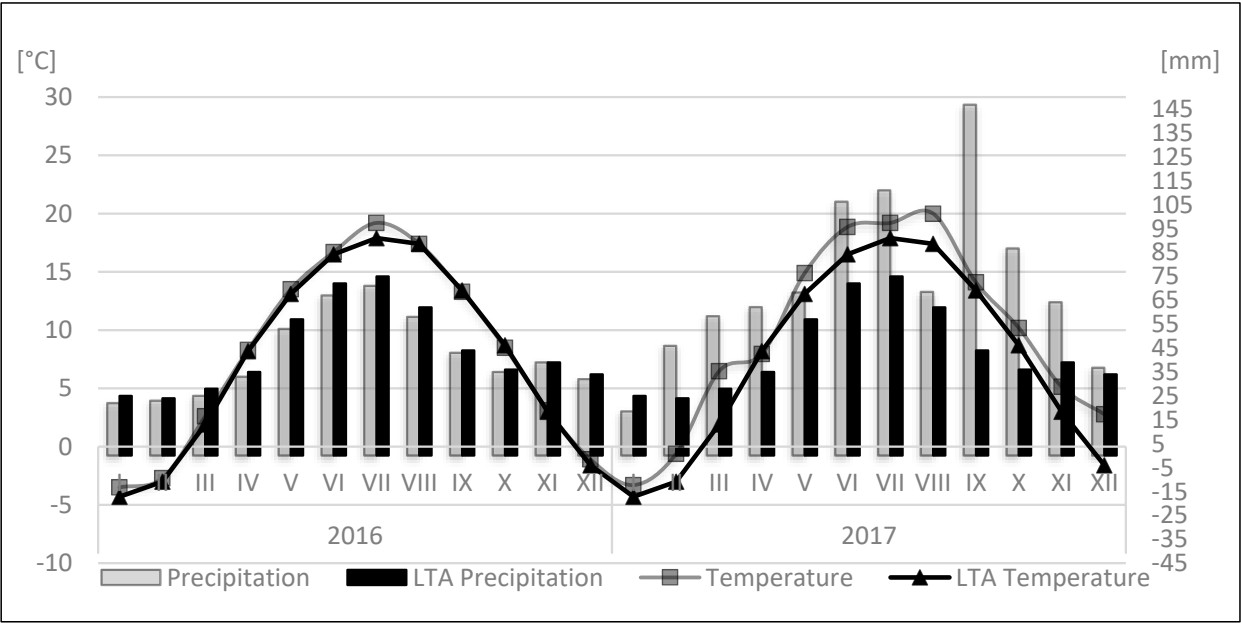

**Figure 1.** Data on average temperatures and rainfall at the experimental site in 2016 and 2017 growing seasons. LTA, the long-term average for years 1982–2012.

## 4. Conclusions

Tested compounds (total ascorbate and total phenolic concentrations, as well as individual phenolic compounds (+)-catechin, (−)-epicatechin, chlorogenic acid, phloridzin, and rutin) were highly tissue type and cultivar dependent. The peel of all tested apple cultivars was characterized by significantly higher antioxidant properties compared to flesh, which confirms its health value and recommendation for consumption. Compared to 'Gala S.', apples of 'Chopin' and 'Granny S.' cultivars expressed much lower skin-to-flesh antioxidant potential differences. The lowest differences between tissue types were in the case of chlorogenic acid and flavan-3-ols, followed by total phenolic compound and ascorbate concentrations. 'Chopin' was definitely distinguished by the highest concentration of ascorbate in the peel and flesh and expressed a higher concentration of flavanols, especially compared to 'Granny S.' In contrast to 'Gala S.', 'Chopin' was richer in chlorogenic acid, (+)-catechin, and (−)-epicatechin in the flesh. A narrower range of differences between the concentration of antioxidants in apple peel and flesh could mean better health-promoting properties. Weather conditions in the season can significantly change the quantitative and/or qualitative characteristics of fruit internal quality. It should be noted that the study characterized a new cultivar 'Chopin' with green skin, resistant to scab, in terms of antioxidant properties. The information can be useful for both producers and consumers,

especially since there are few green-skinned apples with higher acidity on the market (most of them are old cultivars with a small cultivation area). Such cultivars may have wider uses than dessert ones. Commercially important cultivars, such as 'Mutsu' and 'Golden Delicious' (green-skinned, also grown in Poland), are classified as sweet. However, the topic can be extended to new compounds, especially phenolic ones, which have been evaluated to a limited extent, as well as to other green-skinned cultivars. The obtained results can be used in the further selection of genotypes resistant to stress and with a high content of biologically active compounds not only in the peel, but also in the whole apple fruit.

**Author Contributions:** Conceptualization, B.Ł. P.L.; methodology, B.Ł.; formal analysis, M.S. and P.L.; investigation, M.S.; data curation, M.S.; writing—original draft preparation, M.S.; writing—review and editing, B.Ł and P.L.; visualization, P.L.; supervision, B.Ł. All authors have read and agreed to the published version of the manuscript.

**Funding:** This research was funded by the Polish Ministry of Science and Higher Education with funds from the Institute of Horticultural Sciences.

**Institutional Review Board Statement:** Not applicable.

**Data Availability Statement:** Not applicable.

**Conflicts of Interest:** The authors declare no conflict of interest.

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
