# Peer review of "Peel to Flesh Bioactive Compounds Ratio Affect Apple Antioxidant Potential and Cultivar Functional Properties"

_agriculture, doi:10.3390/agriculture13020478_

Round 1

Reviewer 1 Report

Thanks for your good study. Comments and questions are mentioned in attached file.

Reviewer 2 Report

The manuscript entitled “Peel to flesh bioactives ratio affect apple antioxidant potential and cultivar functional properties: two years study” deals with the concentration of antioxidants in peel and flesh of different apple cultivars in terms of their health-promoting properties.

The topic is interesting and up-to-date, fitting to the “Agriculture” scope, but the novelty of the study cannot be seen clearly in the text. Since cultivar ‘Chopin’ is new one, the discussion should focus more on this cultivar. Accordingly, the question arises, why did you obtain ‘Granny Smith’ from the market? Didn’t you have another green cultivar available at the experimental orchard? Thus, for comparison you would have a cultivar that was grown in similar conditions, and it is known, as you claim too, that growing conditions affect the content of bioactive substances.

As well, the major issue is statistical analysis of the average values. Why did you separate the peel from the flesh, did all of the analyses, but did not process them statistically? The selection of cultivars with different skin colors makes sense only if the focus is on the results obtained from different tissues, which would make these research significant.

There are also some specific comments:

Line 25: “Two methods of estimating the total phenolics concentration were involved.” Why is this important? If it is, then both methods should be specified.

In the study there is a terminological issue; terms that are not commonly used in the scientific literature, such as “bioactives”, “free radicals neutralization” should be avoided.

Line 57: “Considering the weight of the fruit,…” did you mean the fresh weight?

The use of 3 methods for the measurement of antioxidant activity and 2 for the total phenolics content should be justified and corelated to the aim of the investigation.

Lines 84-89: these sentences should be moved to M&M section, subsection 2.1 Plant material

Line 119: “The results were compared with the calibration curve” Calibration curve is used for the calculation, but not for comparison

For determination of total phenolics concentration, range of standard concentrations should be specified.

Table 1: “AB AC BC” should stand “A×B A×C B×C” instead? Also, consider to change “ni” to “ns-not significant”

Did you manage to detect more than 5 mentioned phenolic compounds by HPLC? Or you presented only those that were dominant in your samples? This should be discussed.

Tables 2, 3 and 4: why there was no statistical analysis for the AP and AF values, but only for AV?

Line 266: “The recommendation is to evaluate antioxidant activity by three different methods” who recommended this and why? If ABTS test can measure both hydrophilic and lipophilic antioxidants, it is convenient and sufficient.

Lines 275-276: “However, in one test (ABTS), no significant differences between the cultivars in antioxidant capacity were observed” would it remain the same for AP and AF values, if they were statistically analyzed?

Table 6: it is not clear what is represented here! “Differences in biologically active compound concentrations (mg kg-1 FW) and total antioxidant capacity…” which total antioxidant capacity???

Conclusions section should be rewritten to emphasize the importance of obtained results in terms of consumers/producers preferences: which cultivar could be recommended for consumption and in which form-peeled or not?  Thus, the title makes sense!

Reviewer 3 Report

The manuscript entitled "Peel to flesh bioactive ratio affect apple antioxidant potential and cultivar functional properties: two years study" presents data regarding secondary metabolites distribution in apple fruit depending on tissue, season, and environmental conditions.  All of the remarks and suggestions are given in the attached PDF file, so when you open it up in PDF reader, you can see all of the comments and suggestions.

The major points to address are:

-selection of the experimental object. As said in the comments, maybe instead of the red cultivar, another Polish green one could fit better. The red one gave expected higher values, so it is not the best choice for comparison.

-use of uncommon terms, even in the title. For instance, bioactives  is not proper one. Bioactive compounds sound more academic...

-style of presentation, especially in R&D section has to be significantly approved. There is a lack of proper discussion of obtained data, there are a lot of generalisations and review-like trivial sentences.

-Tables should be rechecked, or captions. Sometimes it is hard to follow their content, as suggested in PDF file. Again, please be sure to open it in PDF viewer in order to see all of the comments.

A major revision, including rewriting of some parts, is needed before it can be published in Agriculture Journal.

Round 2

Reviewer 2 Report

The revised version of the manuscript entitled "Peel to flesh bioactive compounds ratio affect apple antioxidant potential and cultivar functional properties" is significantly improved, but it still needs some interventions. It seems like the revision was done in a hurry, so new issues emerged. Together with my comments, they could be seen in the attached pdf file.

Therefore, authors are recommended to check the entire text carefully before resubmitting it.

Author Response

Dear Reviewer,

thank you for all your additional remarks. We appreciate them very much. All issues raised in the second review were corrected and included in the manuscript.

Sincerely,

Authors

Reviewer 3 Report

The authors responded to every question raised in a satisfactory manner. The manuscript looks more informative and proper statistics significantly improved its quality. But, I would kindly suggest the Authors to give one more critical overview of the overall soundness and clarity of the manuscript and to resolve some new ,,grey areas" in the changed text. After this, final self-checking, the manuscript can be published in the Agriculture journal.

Also, a part of the Authors' responses to my comments can nicely fit in to the manuscript, maybe in Introduction. In relation to the selection of experimental object we would like to add, the main aim of the study was to characterize the new cultivar ’Chopin’ with green skin and higher fruit acidity, resistant to biotic stresses.  We hypothesized that due to the high resistance of this variety to biotic stresses, it will have a high antioxidant potential and it would be interesting to compare it with the red-skinned variety, also economically important. There are few green-skin apple cultivars, most of them are old cultivars with a small cultivation area. In fact, the antioxidant potential of the tested green-skinned apple varieties did not differ so much from the red-skinned Gala S. 

Author Response

Dear Reviewer,

thank you for all the additional comments and corrections. We appreciate them very much. All the issues raised in the second review were corrected and incorporated in the manuscript.

Sincerely

Authors